# Aquaphotomics Research of Cold Stress in Soybean Cultivars with Different Stress Tolerance Ability: Early Detection of Cold Stress Response

**DOI:** 10.3390/molecules27030744

**Published:** 2022-01-24

**Authors:** Jelena Muncan, Balasooriya Mudiyanselage Siriwijaya Jinendra, Shinichiro Kuroki, Roumiana Tsenkova

**Affiliations:** 1Aquaphotomics Research Department, Graduate School of Agricultural Science, Kobe University, Kobe 657-8501, Japan; jmuncan@people.kobe-u.ac.jp; 2Department of Agricultural Engineering, Faculty of Agriculture, University of Ruhuna, Mapalana 81100, Sri Lanka; jinendra@agri.ruh.ac.lk; 3Laboratory for Information Engineering of Bioproduction, Graduate School of Agricultural Science, Kobe University, Kobe 657-8501, Japan; skuroki@dragon.kobe-u.ac.jp

**Keywords:** cold stress, stress tolerance, soybean, water, near infrared spectroscopy, aquaphotomics, water molecular species

## Abstract

The development of non-destructive methods for early detection of cold stress of plants and the identification of cold-tolerant cultivars is highly needed in crop breeding programs. Current methods are either destructive, time-consuming or imprecise. In this study, soybean leaves’ spectra were acquired in the near infrared (NIR) range (588–1025 nm) from five cultivars genetically engineered to have different levels of cold stress tolerance. The spectra were acquired at the optimal growing temperature 27 °C and when the temperature was decreased to 22 °C. In this paper, we report the results of the aquaphotomics analysis performed with the objective of understanding the role of the water molecular system in the early cold stress response of all cultivars. The raw spectra and the results of Principal Component Analysis, Soft Independent Modeling of Class Analogies and aquagrams showed consistent evidence of huge differences in the NIR spectral profiles of all cultivars under normal and mild cold stress conditions. The SIMCA discrimination between the plants before and after stress was achieved with 100% accuracy. The interpretation of spectral patterns before and after cold stress revealed major changes in the water molecular structure of the soybean leaves, altered carbohydrate and oxidative metabolism. Specific water molecular structures in the leaves of soybean cultivars were found to be highly sensitive to the temperature, showing their crucial role in the cold stress response. The results also indicated the existence of differences in the cold stress response of different cultivars, which will be a topic of further research.

## 1. Introduction

Soybean (*Glycine max* (L.) Merr.) is one of the most important crops in the legume family with significant economic importance. It is a highly valued food in human and animal diet [1,2] and has important medicinal and industrial applications [2,3]. Soybean plants are susceptible to cold stress: cold halts the growth or results in injuries during all stages of development [4,5,6,7,8,9]. Despite these constraints, soybean has continued its expansion into cool climatic areas of the world [10,11]. In such areas, plants often undergo several degrees of low-temperature stress, and occasional cold stress injuries lead to decreased crop productivity and significant economic losses [12].

Soybean quality and production are dramatically affected by various abiotic stresses and a thorough understanding of the plant stress response is important for developing and breeding soybean with improved stress tolerance ability. Plants respond to all abiotic stresses with a series of morphological, physiological, cellular, biochemical and molecular changes [13]. Their purpose is the adaptation to the existing stress conditions and counteracting stress effects [14]. Cold stress, defined as the temperature in a range low enough to suppress growth without ceasing cellular functions, is known to induce several abnormalities at various levels of cellular organization [15]: (1) altered fluidity and damage of the membranes [16]; (2) the decrease in the uptake of nutrients and water, leading to cell desiccation and starvation [17]; (3) the conformational changes of proteins and nucleic acids [9]; (4) the decline in the rate of metabolic processes, reframing of gene expression [18] and reduced cellular respiration [19]; (5) accumulation of osmolytes and cryoprotectants [20] and (6) generation of reactive oxygen species [9,19,21].

The ability to measure plant stress responses in vivo is becoming increasingly important and methods are sought for rapid assessment of the stress response. Therefore, the development of non-destructive, rapid methodologies for early detection of plant response to cold stress during its growth, on the spot, is of high importance for both development of new varieties and as feedback in the agricultural industry.

Short-wavelength near-infrared (NIR) spectroscopy is a promising technique for fast and non-destructive analysis of biological materials. This region, called the “optical window”, is the most useful region in the NIR for analyzing biological samples since it allows deeper penetration and non-destructive measurements. The acquired NIR spectra allow simultaneous analysis of many biomolecules in vivo. The absorption of molecules in the NIR region is due to the combinations and overtones of vibrations such as stretching and bending of CH, –OH and –NH functional groups, which engage in hydrogen bonding [22]. These functional groups are the primary structural components of major plant compounds—water, proteins, oils and starch [23,24,25]. Compared to the water content, the rest of the plant compounds are present in small quantities, resulting in their low signal in the NIR region—their absorbance bands are often overpowered by water absorption. However, water, with its strong capacity for hydrogen bonding, is very sensitive to any compositional or environmental changes that a biosystem experiences, which in turn produces differences in its spectrum, making it a source of information about the system as a whole and its current environmental conditions [26]. This property of water—that in an interaction with light it behaves like a mirror, revealing the structure (and function) of the system as a whole—is the basis of the aquaphotomics method and scientific discipline [26]. It extends the possibilities of traditional spectroscopy and offers a novel tool for studying biological systems [27].

Many forms of biotic stress, such as viruses [28], and abiotic stress, such as cold, drought, or salinity [29], affect water behavior on a cellular as well as on the whole plant level, which provides the rationale to apply aquaphotomics to study the stress response. The overall performance of a plant towards cold stress is a complex molecular phenomenon [30] strongly linked to the water response at the molecular structure level. The usefulness of aquaphotomics NIR spectroscopy was already demonstrated for non-destructive detection of early response to biotic stress in virus-inoculated soybean plants 2 weeks prior to the appearance of visual symptoms [28]. That work was the first to report evidence of a considerable impact from a virus infection on the hydrogen bonding network of water molecules in the infected soybean leaves and to suggest that reorganization of water at the molecular level is a part of a plant’s response to stress conditions. The subsequent research on abiotic stress, specifically, desiccation stress and differences in response between resurrection plants (extremely desiccation-tolerant) and non-resurrection plants, has also provided significant new insights into the importance of the molecular structure of water for the preservation of plant tissues and survival in stressful conditions [31].

This paper reports the results obtained using a portable, non-destructive NIR instrument for detection of cold stress response in leaves of different soybean cultivars genetically modified to have different tolerances to cold stress. Using an aquaphotomics approach to NIR spectral analysis, we specifically aimed at achieving the following objectives: (1) obtaining the NIR spectral signature of cold stress in soybean cultivars’ leaves that can serve as a tool for early stress detection, and (2) better understanding the physiological role of water molecular species in a cold stress response.

## 2. Results

### 2.1. Raw Absorbance Spectra of Stressed and Non-Stressed Soybean Plants

The mean leaf absorbance spectra (LogT^−1^, where T is leaf transflectance) for soybean cultivar varieties grown continually for three weeks at 27 °C (from now on, referred to as “normal” or “no stress” conditions) and those grown for two weeks at 27 °C and then for one week at 22 °C (from now on, referred to as “cold stress” conditions) are plotted in Figure 1. The main feature of these spectra is a large absorbance peak in the visible region between 650 and 660 nm. This spectral feature is related to the light absorption by chlorophylls in the soybean leaves, which occurs in the visible part of the spectrum; the largest amount of energy is absorbed by chlorophyll *a* around 660 nm [32,33,34]. In the near-infrared domain (700–1050 nm), except for the differences in baseline, such strong spectral features are not visible.

The mean absorbance spectra of soybean leaves before the imposed temperature stress showed a small difference in the baseline offset between the cultivars, with the highest baseline spectral profile belonging to the most susceptible cultivar E (cyan solid line, Figure 1a, inset) and the lowest to the least susceptible cultivar A (black solid line, Figure 1a, inset). However, when the temperature was decreased to 22 °C, the mean absorbance spectra of soybean leaves showed decreased absorbance over the entire region. Comparison of the spectra averaged for all the plants grown at 27 °C and for all of the plants exposed to a 5 °C decrease in temperature (Figure 1b), revealed that decreased absorbance in the entire range is, on average, a common spectral behavior of all cultivars.

From the different spectra calculated for each cultivar, by subtracting the average spectrum of the plants grown in stressed conditions from the average spectrum of the plants grown in normal conditions (Figure 1c), we also observed that this decrease is the least intensive in the most cold-tolerant cultivar A and the most intensive in the cold susceptible cultivar E. Interestingly, only cultivar A displayed the unique feature of an actual subtle increase in absorbance, in the area 870 to 890 nm (approximately around 872 nm), which is usually attributed to the band of carbohydrates [35].

The near infrared part of the spectra when enlarged (~700–1050 nm, insets in Figure 1a–c) shows a strong baseline offset caused by light scattering (which increases the effective pathlength) or other physical differences in thickness or anatomy of the leaves. Despite this, subtle nuances in the shape of the spectral lines suggest that spectral differences also arise from the structural changes in the components of the leaves, which is especially noticeable in difference spectra in Figure 1c at the indicated wavelengths 782, 815, 872, 944 and 998 nm. In the analyzed range, both the second and thrid vibrational frequency overtones of the water OH stretching vibrations are located: second overtone around 970 nm and the third around 738 nm [36,37,38]; also around 836 nm is the third overtone of the combination band [39]. Since in the normal conditions the relative water content of soybean leaves is around 90% [40], it can be assumed that changes in the spectra of leaves in this region would predominantly originate from the water absorbance bands. Water is a strong absorber of NIR light and the spectra of samples with high water contents (>80%) are strongly dominated by the signature from water [41].

The changes in baseline might occur due to several reasons. First, the thickness of the leaves, which gives rise to different optical pathlengths is closely related to the water content of the leaves, as it was reported not only for soybean, but other plants as well [42]. There are reports connecting the cold stress in soybean with a decrease in the relative water content in leaves but at temperatures lower than employed in this research [43]. Furthermore, a horizontal shift in part of the 950–970 nm region was reported to be related to changes in plant leaf water status during water stress [44,45].

In order to analyze all the spectral changes in more depth, in the following analysis, the baseline shift and slope were removed using adequate preprocessing techniques.

### 2.2. Principal Component Analysis (PCA)—Exploratory Analysis of Cold Stress Effects on Spectra of Soybean Cultivars’ Leaves

In order to better examine the changes in leaves due to the imposed stress and enhance the absorption effects in the spectra, further analysis was performed on the truncated region 780–1000 nm, excluding the part attributed to pigments that may dominate the analysis.

The results of PCA, presented as scores and loadings plots in Figure 2 helped in the detection of patterns in the spectral behavior of examined leaves. The scores plot for the first three principal components (which together described 97% of the variance in the spectra) showed sharp separation in two large clusters along the direction of PC1 (Figure 2a). On closer inspection, it was revealed that the PC1 component (which explained 81.3% of total variance) separated the group of non-stressed plants located in the negative part of PC1 from the group of plants exposed to cold stress located in the positive part. The loadings of principal components showed the importance of variables (wavelengths) for the computation of each of the PCs. The loading of PC1, which was the most important for separation of the plants in no-stress and stress conditions, showed several important features: positive peaks at 815, 926, 944 nm (specific for the plants during cold stress conditions) and negative peaks at 873 and 985 nm (specific for the plants in the absence of cold stress) (Figure 2a). The bands 926, 944 and 985 nm, being located in the second overtone of water, can be attributed to absorbance of different water molecular species. Specifically, 926 nm can be attributed to the bands of proton hydrates [46,47] or water hydration shell [48], 944 nm to free water molecules [49,50], while 985 nm to hydrogen-bonded water [51]. The band at 926 nm can alternately be assigned to lipids; around 930 nm in biological samples there is usually a characteristic small lipid peak [52,53,54] to the lipid–water mixture [55], though we cannot exclude the water–lipid interaction spectral feature. Bands at 815 nm and 873 nm may be attributed to carbohydrates, although in different forms, soluble carbohydrates and starch, respectively. The more detailed assignments will be provided later in the Discussion.

Next, in the PC1-PC2 scores plot, it can be seen that PC2 separates cold-susceptible cultivar E in the non-stressed conditions—its scores are located in the negative part of PC2, in contrast to all others. The loading vector of PC2 shows positive peaks at 831 and 900 nm, while in the negative part there is a large, broad peak around 869 nm and a smaller one at 961 nm. Further, in the PC1-PC3 scores plot, PC3 separates cold-tolerant cultivar A during stress conditions from all other cultivars—scores of this cultivar are located in the positive part of PC3. The loading vector of PC3 shows positive peaks at 813, 876, 890, 908, 922, 942 and 995 nm and negative at 835, 855, 961 and 979 nm.

In summary, PCA analysis showed sharp separation between the spectra of plant leaves from the investigated soybean cultivars when they were grown at optimal temperature and after the temperature decrease. Further, the information from lower-order PC components showed the existence of differences between cultivars, particularly separating the weakest cultivar E during normal conditions, while the strongest cultivar A showed a marked difference during stress conditions.

### 2.3. Soft Independent Modeling of Class Analogies for Detection of Plants’ Response to Cold Stress—Discrimination of Non-Stressed and Stressed Plants

Soft independent modeling of class analogies (SIMCA) was first applied with the purpose of supervised classification of plant leaves’ spectra according to the conditions at which they were grown, i.e., no stress and cold stress, in order to develop a model for cold stress detection.

The classification accuracy for the test set was 100%, while the interclass distance (Mahalanobis distance) was 4.53, which shows reliable, strong separation between the classes [56,57] (Figure 3a). Interestingly, on Cooman’s plot (Figure 3a) in the class of cold stress, a separation was detected within the class-cultivar A (the most cold-tolerant) in which it appeared separated from the others, further supporting the results of the PCA analysis in that there is a difference in the reaction of this cultivar to cold stress when compared to the rest of the cultivars.

To further explore this finding, SIMCA analysis was repeated in the same way as before, but after the spectra of cultivar A were excluded from the dataset. The classification accuracy in this case was also 100%, while the interclass distance increased to 6.67 (Figure 3b). In this case, the class seemed well-defined without any cultivars standing out. The increase in interclass distance indicates that the exclusion of cultivar A actually influences better separation of the classes of normal and cold stress conditions, as if the difference in cultivar A before and after cold stress is not so big. It is interesting to note that, in both analyses, plants in the normal conditions, in general, showed more variations within the class (scattering of scores can be observed in Figure 3a,b), in contrast to the cold stress class scores.

The discriminating powers of both SIMCA analyses were investigated for the wavelengths in the NIR spectra with the highest contribution for the distinction between classes (not stressed vs. stressed plants) (Figure 3c). The most significant wavelengths (highest discriminating power) in the case of the first SIMCA analysis, performed on the whole dataset, were observed at 799–03, 827, 868–874, 880, 900, 908, 918–922, 928, 934, 943–946, 959, 973, 985 and 995–996 nm. When the spectra of cultivar A were excluded, the discriminating power lacked a peak at 827 nm, showing this absorbance band is specifically important for separation of other cultivars and A cultivar; it lacks the discriminating power when B, C, D and E cultivar are being compared. The peak at 868–880 nm also changed, adding much more weight to the band at 880 nm and making it the most influential variable for discrimination of cultivars B, C, D and E.

### 2.4. Aquagrams

Because this part of the NIR spectra contains numerous overlapping bands it was deemed necessary to examine in more detail the nature of absorbance bands and how their assignments and interpretation can be related to what was already observed during the analysis. Therefore, the first step of the analysis was to calculate aquagrams in order to present the differences in the stress response of all cultivars together as spectral patterns to find the general features of the cold stress response in soybean (Figure 4).

The aquagram was calculated over the entire spectral region to indicate the regions of importance for separation between the cultivars grown in normal and temperature stress conditions, from the aspect of main leaf tissue chromophores.

The aquagram indicated six regions that show a marked difference in leaves after the cold stress, which can be interpreted as follows:Region 772–799 nm: part of the third overtone of water stretching vibrations encompassing bands of hydrogen-bonded or ice-like water [36,58,59] found to be highly correlated to the sample temperature [59,60]. The temperature stress resulted in a marked decrease in absorbance.Region 800–830 nm: absorbance region that we tentatively assigned to CH of carbohydrates or hydrocarbons, not excluding their cumulative effect on the water molecular structure. The literature sources report the following absorbance bands and their interpretation in this range: 810 nm—related to oxidative metabolism in various cell types and cell proliferation [3,7,8,9,10,11], 815 nm—related to oxidation and the state of chloroplasts [61], 813 nm—absorbance band of aliphatic hydrocarbons [58,62], such as ethylene—a plant hormone with a role in the regulation of oxidative stress [63], shown to be produced during temperature stress in soybean leading to the oxidative injury [64]. The region also contains absorbance bands that may be attributed to water; specifically, 827–830 nm can be an absorbance band of small protonated clusters [46,51,59,65,66], while 814–816 nm a protein–water interaction (unpublished data) or carbohydrate–water interaction ([67], unpublished data). The temperature stress resulted in a marked increase in absorbance in this region.Region 830–840 nm: Absorbing region of both carbohydrates and water, with water being a stronger absorber [35]. Centered at 836 nm is the second overtone of the combination band of water [39]. According to numerous sources, the 835–841 nm can be attributed to water highly influenced by temperature [59,60,68,69,70,71]. Several absorbance bands of small proton hydrates are identified within this region: at 837 nm-(+H(H_2_O), +H(H_2_O)_2_), +H(H_2_O)_4_, +H(H_2_O)_6_ [46,51,59,65,66] and at 841–841.5 nm-(+H(H_2_O), +H(H_2_O)_2_), +H(H_2_O)_4_, (H+·(H_2_O)_5_) +H(H_2_O)_6_ [46,51,59,65,66]. The absorbance at wavelength 840 nm was found to be related to the sample pathlength [60].Region 841–900 nm: In this region both water and carbohydrates absorb, but at 870–890 nm is a strong absorbance region of carbohydrates [35,72], in particular the band 878 nm can be attributed to starch [72], major component of the leaves, and one of the key molecules mediating plant responses to abiotic stress, reported to decrease in response to abiotic stress independently of plant species [73]. The absorbance in this region shows a decrease in response to imposed temperature stress.Region 900–959 nm: second overtone of water, the region that can be attributed to various water molecular species that are not involved in hydrogen bonding, i.e., less hydrogen-bonded water. The literature sources show rich information on particular absorbance bands corresponding to the specific water molecular conformations, which can all be connected to their respective locations in the first overtone region (1350–1439 nm), encompassing C1 to C6 Water Matrix Coordinates—WAMACs, that is, water solvation shells, proton hydrates, water vapor, trapped water, free water molecules and the hydration band [26,74,75]. The aquagram shows increased absorbance in this region after temperature stress, which is consistent with our previous findings of biotic stress [28].Region 960–1000 nm: Second overtone of water, the region that can be attributed to various water molecular species that participate in hydrogen bonding, i.e., hydrogen-bonded water. Similar to the previous region, this one can be related to the WAMACs C7 to C11 in the first overtone of water, that is: water dimers, water solvation shells, physi-adsorbed water or bulk water, and water molecules with 2, 3 and 4 hydrogen bonds [26,74]. The aquagram shows decreased absorbance in this region after temperature stress (with the exception of a very small increase at 995 nm) in agreement with what was also observed in region 1, from the 3rd overtone of the same absorbance bands.

The aquagrams show that the absorbance spectral pattern of soybean leaves after the imposed low-temperature stress is vastly different compared to the optimal growing temperature (represented by zero line on aquagram). The difference can be related to the known and reported stress responses in soybean: changes in the water status and water molecular structure reorganization, changes in oxidative metabolism, possibly related to the ethylene hormone and the state of the chloroplasts, increased moisture (gas phase) in the leaves and their thickness and decrease in starch content. However, most importantly, the absorbance bands of water that literature sources indicate as highly influenced by temperature, were found to be the strong signature of the plant leaves’ spectral changes as a reaction to temperature stress, showing direct relationship with the influence of the environment on the water metabolism. The aquagram, Figure 4, testifies about the change in the interaction of light energy and leaf tissues in soybean as a consequence of temperature stress and can serve as a quick visualizing tool for the occurrence of a stress response.

Lastly, retaining only the absorbance bands that showed high importance in the previous analyses, a simple aquagram is made using 12 absorbance bands as radial axes (Figure 5).

The absorbance bands where increased absorbance of soybean leaves occurs in response to cold stress are located at 815, 827, 900, 908, 928 and 944 nm. This absorbance pattern speaks of increased solute accumulation (815, 827 nm [58,59,76,77,78,79,80]), increased interaction of water and these solutes (900, 908 and 928 nm [26,46,48,75,81]) and increased amount of free water molecules (944 nm [26,49,50]). All three phenomena are well-known to occur in plants subjected to stress. While at this point it is not possible to clearly identify what particular solutes are involved, it does not diminish the diagnostic value of the aquagram to clearly show if the plants are under stress or not. Using only 12 wavelengths to form a diagnostic marker also creates a good basis for the development of simply designed and low-cost portable sensors for applications in the field.

## 3. Discussion

In this paper, we used aquaphotomics based on a near infrared spectroscopy method for the evaluation of soybean plants in situ during imposed low-level temperature stress. The research was directed toward two goals: (1) early temperature stress detection in soybean based on the NIR spectral signature of cultivars’ leaves, and (2) better understanding the molecular structure of water in leaves and how it is related to the overall stress response.

All our results, starting from the raw spectra analysis, over PCA exploration, SIMCA discrimination and visual representation using aquagrams, consistently showed strong evidence that a decrease of only 5 °C from the optimal growing temperature produced a response in plant tissues that was captured in the spectra. This decrease could be considered very mild cold stress, indicating that our method captured an early response to the changes in the environment of plants. Previous studies on living systems revealed that even such a small temperature decrease or increase is enough to induce antioxidant activities [82], which means that the method employed in this work is sensitive enough to capture the plants’ stress response early, before the damage of the tissues. It is interesting to notice that detection of cold stress and discrimination analysis produced much poorer results when the whole vis-NIR region was used in the analysis (data not shown), indicating superior power of NIR spectroscopy because of the mild water absorbance used here as a source of information.

In the raw spectra the change in environmental temperature brought a downward shift of the baseline, caused by both physical and chemical changes in the leaves. Even in this initial step of analysis, the differences in cold stress response between the cultivars started to appear. The least cold-tolerant cultivar E showed the most intense change in the spectral profile, while the most cold-tolerant cultivar A, the smallest change, being most stable to the environmental perturbation.

The PCA analysis performed on preprocessed data, from which the effects of physical differences were removed, showed that temperature change is the cause of the largest variation in the spectral data, separating into two distinct groups scores of plants grown at optimal temperature, from the scores of the same plants after the temperature stress. In addition, lower-order PCA components showed distinctive characteristics of cold-susceptible cultivar E, whose scores were separated from the rest of the cultivars even in the conditions of optimal growing temperature, while the characteristics of cold-tolerant cultivar A became more distinguished after the exposure to lower temperatures. These findings indicated that the NIR spectra contain information not only about the stress response but also shed light on the specifics of cultivars and their individual stress responses.

SIMCA analysis further confirmed the results of PCA analysis, allowing persuasive discrimination of classes of plants grown at normal temperature and after they were exposed to cold stress. The value of interclass distance was large enough to confirm distinctive spectral features that characterize the plants before and after the cold stress. The accuracy of discrimination between the plants’ growth at optimal temperature and after the temperature decreased for only 5 °C testifies to the high sensitivity of NIR spectroscopy and the power of utilizing the spectral pattern as a multidimensional biomarker for capturing the systematic stress response. Here, in this analysis, cultivar A was also distinguished compared to others when the stress occurred, once more indicating fewer variations in the less susceptible plant to be its specific stress response. The difference in discriminating power of SIMCA analyses performed with and without cultivar A in the dataset showed a particular absorbance band located at 827 nm to be the feature that is highly specific for the most cold-tolerant cultivar in relation to others.

By presenting the spectral profiles in aquagrams, the general, average reaction of all cultivars showed visually clear distinction of plants’ leaves before and after cold stress. The observed spectral differences were large in six wavelength regions indicated in the aquagram, each of which was related to the main absorbers in leaf tissues and their involvement in the stress response and the wavelength assignment was supported by the current scientific literature. Throughout the analysis, we could witness the consistently repeating several absorbance bands, all of which belong to the indicated six regions found to be related mainly either to absorbance of water or carbohydrates. In summary, the findings based on the aquagram profile show that the early cold stress response is characterized by a decrease in the absorbance of hydrogen-bonded water and starch and increase in weakly hydrogen-bonded water, free water and moisture. It is interesting to notice that water absorbance bands of trapped water and strongly bound water, which are reported as a water spectral pattern related to dehydration and damage of biological tissues [27,74,83], were not observed during this study, indicating that at this stage of cold stress, the leaves did not suffer injuries.

Further on, the changes in absorbance profile were connected to changes in starch metabolism, oxidative metabolism and possibly other indicators of plant stress, such as plant hormones and state of chloroplasts. Starch metabolism has recently emerged as a key determinant of plant fitness under adverse conditions and is well-documented in various plant species but still with fragmentary knowledge about the roles in the stress response [73]. There are indications of its involvement in sugar metabolism enzymes, with differences between soybean genotypes, particularly with respect to starch characteristics in chloroplast ultrastructure [84]. The equal importance of the role of water and starch in the spectral pattern of cold stress in soybean places a new spotlight on the water functionality in the stress response. In particular, the interesting finding of the study is that specific water species, which literature sources describe as highly related to the sample temperature, were found as an important part of the stress response. The studies have reported that plants have the ability to, for example, cool their leaves below that of the temperature of the environment [85]. The temperatures of plants show that even with a 10 °C difference compared to the air temperature, physiological measurements correlated highly with a plant water stress index [86,87]. Plants have the ability to be cooled by transpiration, and in reverse when the stomata are closed, the plant temperature increases. The existing thermal imaging studies have shown the significant effects of cold and water stress on thermal infrared spectra of different plants [88]. Similarly, infrared thermography had been utilized for screening Arabidopsis plants with altered stomatal responses to drought based on the leaf temperature because their leaves appeared colder compared to the wild type [89].

In our research, the result that soybean plants in response to cold stress decrease the absorbance of hydrogen-bonded water and increase the absorbance of weakly hydrogen-bonded water (which is also evident from the changes in the baseline offset) show that plants work against cold environment by “internal heating”, i.e., reorganizing the water molecular structure to be as if the temperature is higher. The observation that the magnitude of changes was highest in cold-susceptible cultivar E while lowest in cold-tolerant cultivar A suggests the possibility that the water in leaves of a cold-tolerant cultivar is already in the more favorable state, which opens up a new line of research direction aimed at examining the relationship between genetic modification and the end-result of a specific water molecular structure in the leaves of cold-tolerant plants.

The existing findings, in addition to the observation of this research, strongly support the possibility that water, not only its content but its structure, has an important role in the regulation of the plant’s internal temperature, and certain water species that are temperature sensitive serve as signaling molecules, possibly initiating a cascade of events in the stress defense system.

As we witnessed in the results of the present research, the genomic base of the soybean cultivars may also dictate how sensitive the water molecular matrix of the leaves is to the perturbation coming from the environment. However, this is a research topic of future physiological, genetic and aquaphotomics studies. Hopefully, the integrated studies of these various omics disciplines may uncover the specifics of the mechanism underlying the different responses among cultivars and result in new strategies to breed soybean for future climates.

## 4. Materials and Methods

### 4.1. Plant Materials

Seeds of five different soybean (*Glycine max*) cultivars that have different degrees of cold stress tolerance, Kitamusume (A), Toyoharuka (B), Toyokomachi (C), Toyomusume (D), Hokkaihadaka (E), with cultivar A being the most cold-tolerant and E being the most cold-sensitive, were obtained from the Tokachi Agricultural Experiment Station (TAES), Hokkaido, Japan. The cold stress tolerance level was based on the Tokachi Agricultural Experiment Station grading based on the field performance. The cold tolerance of the studied cultivars, their cold tolerant index (CTI), was assigned based on the multiple field experiments. There are several research studies that have investigated the cold tolerance ability and field performance of the soybean cultivars studied in this work, and other cultivars as well, which can be used as a source of more information about the genotypic differences [10,11,90,91,92,93].

### 4.2. Experimental Protocol—Experimental Conditions for Cold Stress Investigation

Two hundred plants, 40 plants per cultivar, were grown in plastic pots at 27 °C, 14 h light/10 h dark (day/night) supply (22,000 Lux) for two weeks. After two weeks, 20 plants from each cultivar were moved to 22 °C phytothron (“cold stress” conditions) and the rest were kept continuously at 27 °C for up to three weeks (“normal” conditions—no stress). The temperature of 27 °C was chosen because it is usually encountered in real field conditions, while the mild cold stress temperature of 22 °C, i.e., decrease of only 5 °C, was chosen because it is still within the optimum temperature window for soybean growth (between 20 and 30 °C, [94]). If further reduced, prolonged exposure to low temperatures (15–10 °C) may induce cultivar cold acclimation attributes [95], which was not an objective of this study.

### 4.3. NIR Spectroscopy Measurements

Measurements were carried out non-destructively on single leaves using a handheld type NIR spectrometer FQA-NIR Gun (Shizuoka Shibuya Seiki, Hamamatsu, Japan). Five transflectance spectra (588 nm up to 1025 nm at 2 nm steps) from the first trifoliate leaves of each plant were acquired using a custom designed reflectance probe as previously described (Figure 6) [28]. Briefly, the modified probe allowed constant measuring conditions for all leaves by preventing the environmental light interference, and a hinge-type bottom plate of the modified probe provided a constant white background for all the measurements. The sampled leaf was kept immobile, and spectral acquisition was performed within few seconds, thus minimally interfering with the leaf functionality.

The acquired leaf reflectance spectra (R) were converted to absorbance spectra (A = logR^−1^). For 5 cultivars with 40 biological replicates per cultivar and 5 acquired spectra from the first trifoliate of each biological replicate, a dataset of 1000 absorbance spectra for the analysis was obtained (500 for the plants grown in normal, and 500 for the plants grown in stressed conditions).

### 4.4. Data Processing and Analysis

Principal component analysis (PCA) [96] of the whole spectral dataset in the spectral range 780–1000 nm was performed after detrend and standard normal variate transformation [97] preprocessing to remove the baseline effects and improve separation of the overlapping bands [98]. PCA was used on the whole dataset—firstly, for the general inspection and removal of outliers based on the values of the Mahalanobis distance [99]. Secondly, the aim of performing PCA was to examine the information present in the dataset and to detect possible existing patterns in data and relationships with variables. PCA transforms the original data into new, orthogonal, variables called principal components (PCs) where only the first few contain most of the useful information. The loadings of PCA indicate the importance of wavelengths for computation of each PC.

Soft independent modeling of class analogies (SIMCA) [100] was applied to build a supervised classification model for discrimination of plants in normal and stressed conditions. A SIMCA model was tested on an independent test set containing 50% of the original dataset, formed by selecting every other spectra (odd numbered spectra were used for calibration, even numbered for validation). The test set used for validation was left completely out during the modeling based on the calibration set.

An aquagram was used to visualize the changes in the water molecular matrix in the leaves of all cultivars together in response to cold stress. Aquagrams were calculated using the method for classic aquagrams [99], according to the following equation:(1)$$A_{\lambda }^{\prime }=\frac{A_{\lambda }^{}-\mu_{\lambda }^{}}{\sigma_{\lambda }}$$

In the equation, $A_{\lambda }^{\prime }$ is a normalized absorbance, which is displayed on the aquagram, and *A_λ_* is the absorbance after detrend [98] and standard normal variate transformation (SNV) [97] performed on spectral data acquired in the normal conditions and cold stress conditions separately. The value *μ_λ_* is the mean, while *σ_λ_* is the standard deviation of all the preprocessed spectra together after separately performed transformations. First, the aquagrams were presented using as the wavelengths λ all the wavelengths from the whole spectral range. This way of representing differences allowed for better identification of major absorbers in the leaves and how they changed in response to stress. Finally, as is usual practice in the aquaphotomics analysis protocol, only the most influential absorbance bands were retained based on the frequency of their occurrence throughout the analysis [98], and the simple aquagram was made using only 12 absorbance bands as radial axes.

The two transformations—detrending and standard normal variate transformations—were performed prior to the aquagram calculation to cancel the differences in baseline effects and were performed separately on data acquired during normal and during stress conditions because the raw spectra showed differences in the baseline effects depending on the conditions.

The transformation of spectral data, data analysis and visualizations were performed using the aquap2 package [101] in R Project statistical software [102]. Spectral subtraction and peak detections in second derivative spectra and difference spectra were performed using OriginPro^®^ version 8.5 (OriginLab Corporation, Northampton, MA, USA). Aquagram calculations were performed using Microsoft Excel 2016 (Microsoft Corp., Redmond, WA, USA).

## 5. Conclusions

This study was conducted using near infrared spectroscopy and aquaphotomics as a novel method for non-destructive detection of the cold stress response.

The research showed that the method was sensitive enough to detect the response of the soybean plants even in very mild cold stress conditions and to discriminate with 100% accuracy the plants grown at optimal growing temperature and plants after they were exposed to the cold stress.

The information in spectra that allowed the successful detection of cold stress response was largely based on the water absorbance bands, testifying to the water structure playing an important part in the primary stress response in soybean plants’ leaves. Additionally, the role of carbohydrates and their interaction with water was found to be strongly associated with the cold stress response.

Our results cast new light on the importance of water in plants’ adaptive response to temperature change and its role in the cold tolerance ability of different soybean cultivars contributing to a better understanding of this phenomenon, while at the same time providing a novel principle for the development of rapid, non-destructive methods for cold stress detection that can be performed in the field.

## Figures and Tables

**Figure 1 molecules-27-00744-f001:**
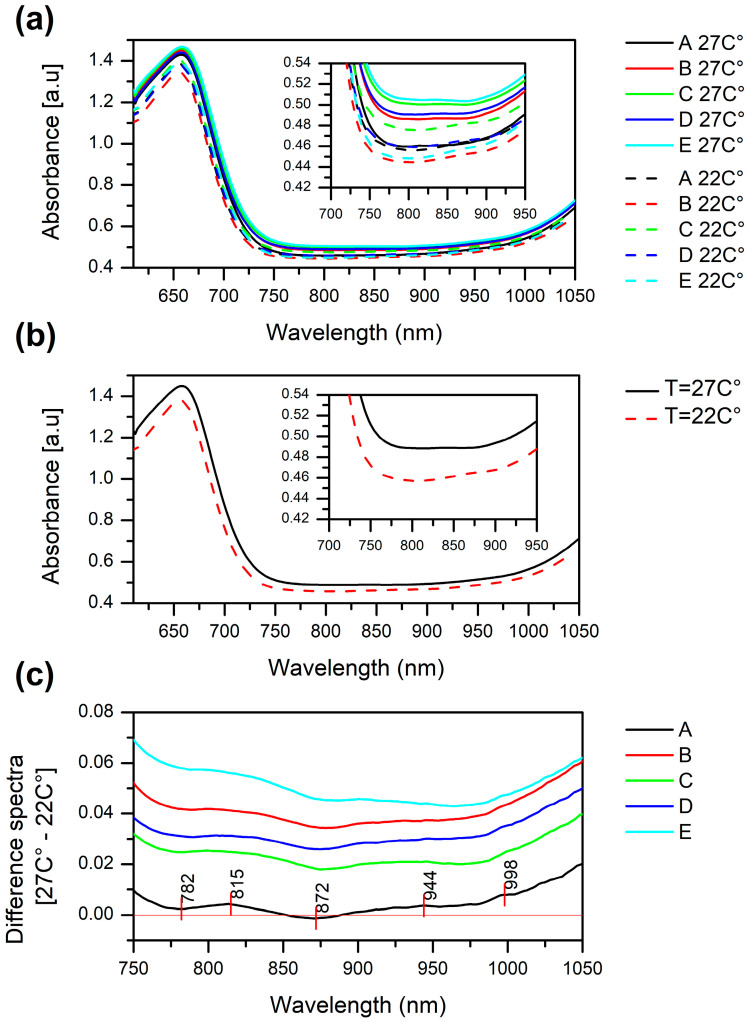
(**a**) Raw absorbance spectra of non-stressed (solid lines) and stressed (dashed lines) soybean plants’ leaves in the vis-NIR region. Averaged for each cultivar separately—in normal growing conditions (27 °C) and in the mild cold stress conditions (22 °C); (**b**) averaged spectra for all cultivars together—in the absence of stress and in the cold stress conditions; (**c**) the difference spectra calculated as the averaged spectrum for each cultivar at 22 °C was subtracted from the average spectrum of the same cultivar at 27 °C.

**Figure 2 molecules-27-00744-f002:**
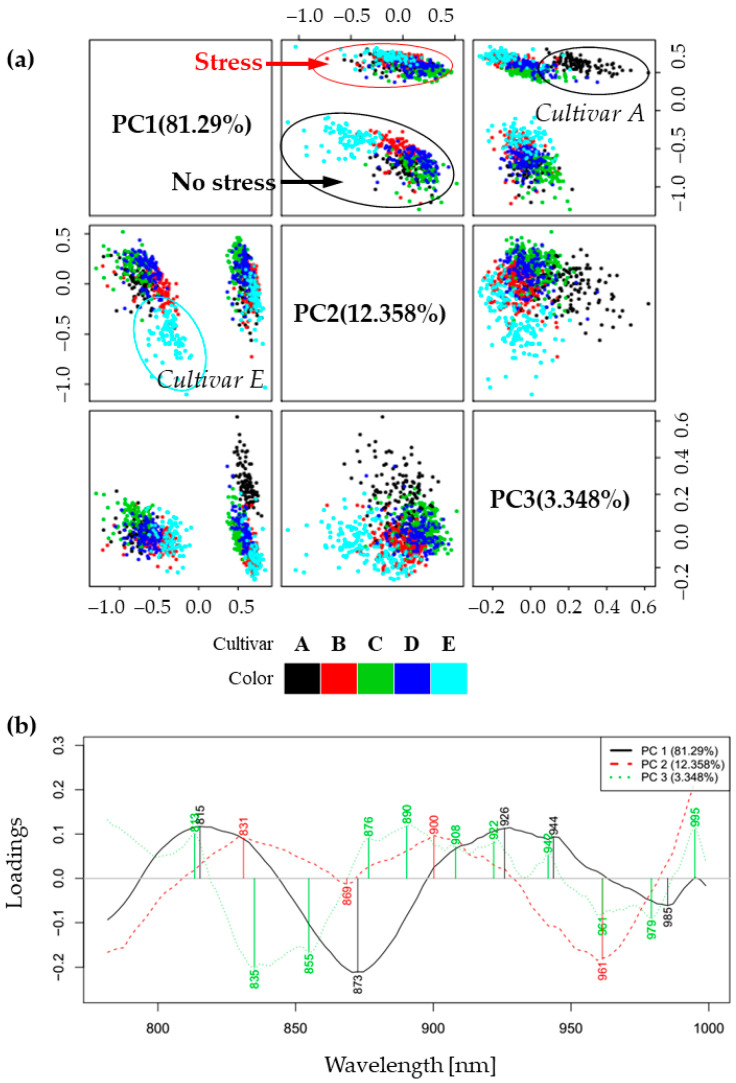
Principal component analysis results: (**a**) Score plots for the first three components revealed separation of plants according to the growing conditions. Two large groups of scores in the PC1-PC2 space correspond to the spectra of plants in the optimal growing conditions and during cold stress. In the score plots PC1-PC3 and PC2-PC3, differences were observed between cultivar A and cultivar E, respectively, compared to the other cultivars; (**b**) loadings of the first three principal components describe 97% variation in the spectra. The loading of the PC1 describes variations in the spectra of plants’ leaves as a result of changes in the temperature of the growing environment, while the PC2 and PC3 loading show spectral characteristics that distinguish cultivar E and cultivar A, respectively, compared to the other cultivars.

**Figure 3 molecules-27-00744-f003:**
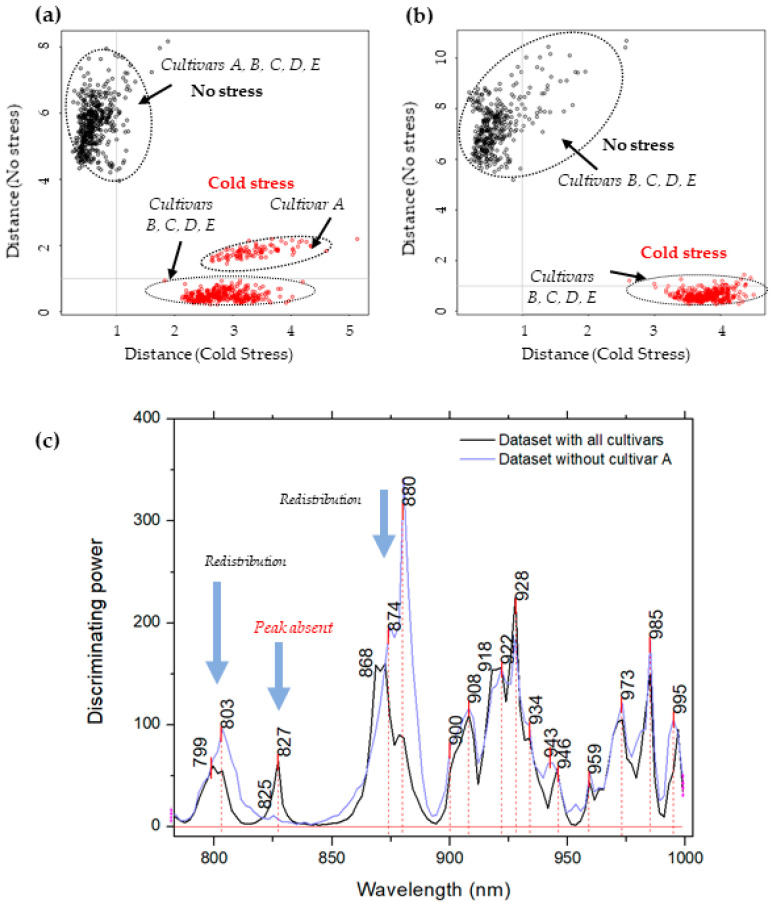
(**a**) SIMCA analysis results when modeling was performed using the dataset with all cultivars (A, B, C, D and E). Cooman’s plot of plants grown in normal conditions (27 °C) (black) and plants exposed to mild cold stress for one week (22 °C) (red) shows excellent separation between plants and reveals distinctive stress response in cultivar A; (**b**) SIMCA analysis results when modeling was performed after the spectra of cultivar A were excluded from the analysis. Cooman’s plot of plants grown in normal conditions (black) and plants exposed to mild cold stress for one week (red) shows strong, reliable separation of classes without distinction of cultivars within the class; (**c**) discriminating powers of SIMCA analyses. Comparison of discriminating powers shows that in both cases almost the same wavelengths contributed to the successful separation of classes of plants grown at normal temperature and in cold stress conditions. The exception is a peak at 827 nm, which is missing in the discriminatory power of SIMCA performed on the dataset without cultivar A.

**Figure 4 molecules-27-00744-f004:**
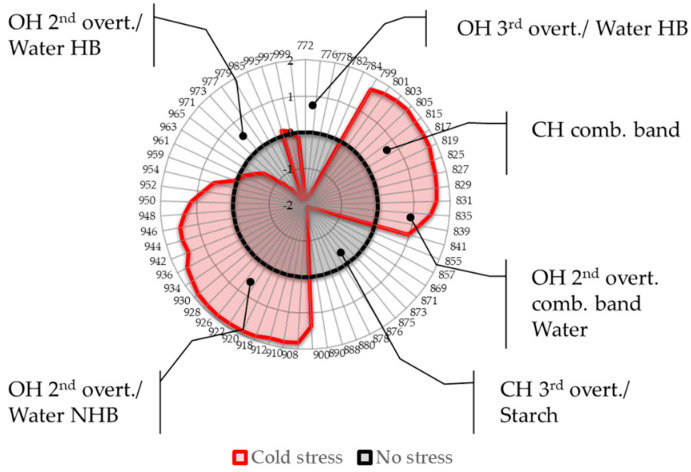
Aquagrams showing differences in the average spectral pattern of all cultivars calculated over the whole spectral range to present general effects of the cold stress response. The main features of the stress response are found in the 2nd overtone of water region, 2nd overtone of water combination band and 2 regions that can be attributed to 3rd overtone and combination bands of CH compounds. (HB—hydrogen bonded, NHB—non-hydrogen bonded).

**Figure 5 molecules-27-00744-f005:**
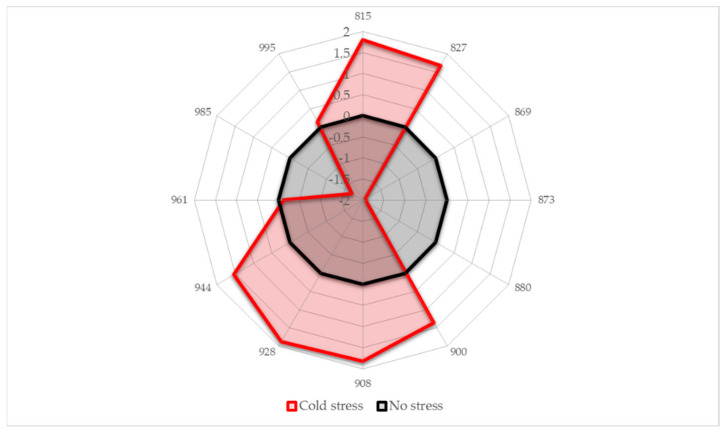
Aquagrams showing differences in the average spectral pattern of all soybean cultivars in the conditions of no stress and during cold stress.

**Figure 6 molecules-27-00744-f006:**
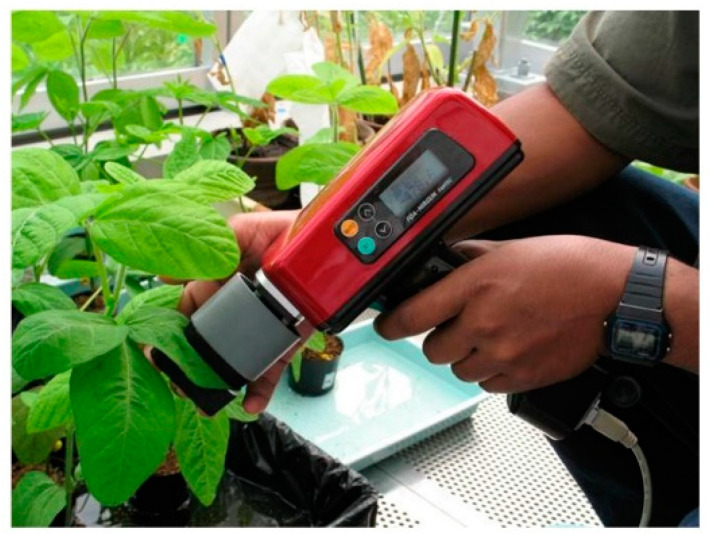
Acquiring soybean leaf spectra by handheld NIR spectrometer with a custom designed probe.

## Data Availability

The data presented in this study are available on request from the corresponding author.

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
