# Peer review of "Aquaphotomics Research of Cold Stress in Soybean Cultivars with Different Stress Tolerance Ability: Early Detection of Cold Stress Response"

_molecules, 2022, doi:10.3390/molecules27030744_

Round 1

Reviewer 1 Report

The paper entitled Aquaphotomics Research of Cold Stress in Soybean Cultivars 2 with Different Stress Tolerance Ability. Early Detection of 3 Cold Stress Response investigates an interesting and valuable work. The work of the authors is very useful and provides a tool for the detection of cold stress of soybean plants.

The article is very well written, having small mistakes only. In general, I think that the article is very valuable, and personally I like that the spectral assignations are given very well and detailed.

However, there are some minor things that could be corrected. Please, find the details below. For better performance and understanding, authors should consider using the same coloring system on the figures.

Fig 1 C, how the bands were selected. The authors mention that they have performed the wavelength picking with the software. What was the bandwidth along this?

Line 161-165 – this part should be mentioned in the Materials and methods part, instead of the results.

In my opinion, figure two should be placed in the PCA section instead of the SIMCA. Moreover, the described variances (in %) could be inserted to the diagonals.

Line 205-210 and 221-222 is again the method itself. Authors should consider giving these details in the materials and methods. Moreover, most of this information was mentioned twice (in the methods, and in the results). Authors should avoid this.

Materials and methods:

            Materials: Is there any publication about the tolerance of the different species published? If yes, it should be at least cited.

Line 473 – I think correctly non-destructively

line 514-516 – I see based on the results that all the wavelengths were used for the interpretation of the aquagrams. However, I think it's worth to try to choose the most contributing wavelengths (based on the loading of PCA and SIMCA in this case) and prepare an aquagrams also using those wls.

line 523 the package is aquap2

Author Response

Dear Reviewer,

We are very grateful for your kind appreciation and endorsement of our work. We have carefully addressed the general recommendations and specific comments made by the Reviewer and revised the paper accordingly. Our point-to-point responses to the specific comments are provided below for easy reference, and all the revisions in the Manuscript are tracked. In addition we also refer in point-to-point replies in which section and/or paragraph specific revision can be found. We hope we have successfully addressed the minor revision requests and that revised paper can now be accepted for publication. Thank you very much for the time, effort and willingness to help us improve the paper.

Best regards,

Authors

Comment1: authors should consider using the same coloring system on the figures.

Response1: Thank you for this advice. We have created the unified coloring system, and all the figures in the revised Manuscript now follow the same color scheme. We also changed the labeling system of individual plots within the same figure to now be a), b) etc. to avoid the possible confusion with the labels for cultivars A, B, C, D and E, as was suggested by second reviewer. Thank you.

Comment2: Fig 1 C, how the bands were selected. The authors mention that they have performed the wavelength picking with the software. What was the bandwidth along this?

Response2: The bands that were displayed on the figure were found using Peak Analyzer function in the OriginPro 8.5 software. Peak finding settings were set to use the method of Local Maximum (in both directions), which is very simple technique, we were not searching for hidden peaks, therefore we did not use any transformation and/or window, bandwidth etc. The peaks were found just as local maxima/minima along the entire range. The method is rather simple, in fact the same peaks could probably be detected without any tools, just by careful inspection, so we omitted this explanation from the Data analysis section. 

Comment3: Line 161-165 – this part should be mentioned in the Materials and methods part, instead of the results.

Response3: Thank you for this comment, yes we agree that this was repetitive. The lines 161-165 were removed, since there is the same information already present in the first sentence of Data analysis section. It was explained that PCA was performed after detrend and standard normal variate transformation.

Comment4: In my opinion, figure two should be placed in the PCA section instead of the SIMCA. Moreover, the described variances (in %) could be inserted to the diagonals.

Response4: Yes, that is correct, thank you. The figure was misplaced due to the text editing. It is correctly placed in the revised manuscript, it is at the end of the PCA section. The figure was also revised, and now it includes the % of described variances for each PC.

Comment5: Line 205-210 and 221-222 is again the method itself. Authors should consider giving these details in the materials and methods. Moreover, most of this information was mentioned twice (in the methods, and in the results). Authors should avoid this.

Response5: Thank you for observing this. Yes, this was a repetition, and we removed those parts from the revised Manuscript, as the same information already exists in Material and methods. Thank you!

Comment6: Materials and methods: Materials: Is there any publication about the tolerance of the different species published? If yes, it should be at least cited.

Response6: The information about the stress tolerance of each of the cultivars used in the study was received from the Tokachi Agricultural Research Station, which developed the cultivars. We stated this in the section 4.1. Plant materials  “The cold stress tolerance level was based on the Tokachi Agricultural Experiment Station grading based on the field performance.” Unfortunately, a single study that compares all of the five cultivars was not performed, but there are several publications where some of the cultivars we used were assessed, together with some others. We added those publications as sources that can be used to learn more about the performance for some of the cultivars, and also about their genotypic differences.  Please find the last sentence in the 4.1 Plant materials “There are several research studies that have investigated the cold tolerance ability and field performance of the soybean cultivars studied in this work, and other cultivars as well, which can be used as a source of more information about the genotypic differences [10,84–88]. “ that cites 5 newly added publications.

 Comment7: Line 473 – I think correctly non-destructively

Response7: Thank you, yes. We added a dash in the revised Manuscript. Please find the first sentence in section 4.3. NIR spectroscopy measurements with the correction made to “non-destructively”.

 Comment8: line 514-516 – I see based on the results that all the wavelengths were used for the interpretation of the aquagrams. However, I think it's worth to try to choose the most contributing wavelengths (based on the loading of PCA and SIMCA in this case) and prepare an aquagrams also using those wls.

Response9: Thank you for this suggestion. We have performed the revision as requested, and added one more aquagram, given in newly added Figure 5 of the revised manuscript.. Only 12 absorbance bands were retained based on the number of their occurrence and importance during previous analyses. We have also performed small revision in the Data analysis section to explain how this aquagram was developed, and after the Figure 5 one paragraph is added discussing the result and the benefits of this way of presentation. We hope this revision will be found satisfying.

Comment9: line 523 the package is aquap2

Response9: Thank you, yes. That was a typo, and we have corrected it in the revised manuscript. Please find the correction in the last paragraph of the section 4.3. Data processing and analysis.

Reviewer 2 Report

It is very interesting to use NIR to monitor the climate stress to soybean plants.  The manuscript is well organized and written, and I think it is suitable for publication in this journal. I have some minor suggestions for the authors. I am very interested in the absorption peak at 827 nm. Since cultivar A is not sensitive to temperature compared to E. Then the genotype should be very different to E, authors could explain this in detail to help us understand this point. Also, the different proteins may help the leaf to have different capacities to interact with water to have different temperature resistibility.  Also, it could be found in the figures that the letters A B C were used for different purposes, and I suggest the authors use lower case letters to represent different plots or cultivars.

Author Response

Dear Reviewer,

Thank you very much for investing the time to read our Manuscript and very constructive comments and ideas. We are also grateful for your encouragement and endorsement of our paper for publication. We will here address here the specific comments, point-by-point, for easy reference.

Comment1: I am very interested in the absorption peak at 827 nm. Since cultivar A is not sensitive to temperature compared to E. Then the genotype should be very different to E, authors could explain this in detail to help us understand this point. Also, the different proteins may help the leaf to have different capacities to interact with water to have different temperature resistibility. 

Response1: Yes, this was one of the most striking and straightforward differences that we observed. Unfortunately the current literature sources do not offer much explanation about the possible assignment and/or related functionality in connection with this absorbance band at 827 nm, which is one of the future directions of this research. In this work we chose to put the focus on the achieved detection of cold stress very early (this is the reason why the title is Aquaphotomics Research of Cold Stress in Soybean Cultivars with Different Stress Tolerance Ability. Early Detection of Cold Stress Response), in different cultivars, which shows consistency of research results and the general applicability of the NIR method for this purpose. But we are in the process of finishing the second paper (Aquaphotomics Research of Cold Stress in Soybean Cultivars with Different Stress Tolerance Ability. Discrimination of Cultivars and the Role of Water Molecular Structure in Stress Tolerance), where we analyzed the same dataset, with the focus exactly on the differences between the cultivars, and how their genotype actually leads to differences in water molecular structure in leaves, and therefore to difference in cold tolerance. Since this paper is aimed at only detection of cold stress, in response to this comment we provided in “Plant materials” section the information that cultivars have different genotype, and we offer 5 literature sources of research studies where the same cultivars were used, and where there is description of there genotypic information, as related to the cold tolerance ability and field performance.  

Comment2: Also, it could be found in the figures that the letters A B C were used for different purposes, and I suggest the authors use lower case letters to represent different plots or cultivars.

Response2: Thank you for this comment. We absolutely agree, and we have corrected all the figures as advised. Please find the correction in the revised Manuscript. All the figures now how low case letters for individual plots, the figure legends and references to plots in the text are revised to reflect this change.  

Once more thank you very much for the endorsement of the paper and interesting comments.

Thank you,

Authors

Reviewer 3 Report

Paper is very interesting as well as methods of analysis of measured data.

I believe that presented method can help to diagnose state of plants during growing and to help start adequate actions to prevent plant damage.

I recommend to accept the paper and I do not have any other comments.

Author Response

Dear Reviewer,

Thank you very much for investing the time to read our Manuscript and provide a review. We are very grateful to learn of your opinion about the paper, it is very encouraging and we hope that our paper will indeed be a step towards non-destructive diagnosis of plants stress and prevention of damage.

Thank you,

Authors